# Cosmo ArduSiPM: An All-in-One Scintillation-Based Particle Detector for Earth and Space Application

**DOI:** 10.3390/s24123836

**Published:** 2024-06-13

**Authors:** Valerio Bocci, Babar Ali, Giacomo Chiodi, Dario Kubler, Francesco Iacoangeli, Lorenza Masi, Luigi Recchia

**Affiliations:** 1INFN Sezione di Roma, 00185 Rome, Italy; giacomo.chiodi@roma1.infn.it (G.C.); francesco.iacoangeli@roma1.infn.it (F.I.); masi.1796721@studenti.uniroma1.it (L.M.); luigi.recchia@roma1.infn.it (L.R.); 2Department of Electronics, Politecnico di Milano, 20133 Milan, Italy; babar.ali@uniroma1.it; 3Department of Astronautical, Electrical, and Energy Engineering (DIAEE), Sapienza Università di Roma, 00184 Rome, Italy; 4Microchip Technology, 20025 Legnano, Italy; dario.kubler@microchip.com; 5Department of Physics, Sapienza Università di Roma, 00185 Rome, Italy

**Keywords:** silicon photomultiplier, SiPM, radiation-tolerant, scintillator, SoC, microcontroller, IoT, CubeSat, space

## Abstract

Thanks to advancements in silicon photomultiplier sensors (SiPMs) and system-on-chip (SoC) technology, our INFN Roma1 group developed ArduSiPM in 2012, the first all-in-one scintillator particle detector in the literature. It used a custom Arduino Due shield to process fast signals, utilizing the Microchip Sam3X8E SoC’s internal peripherals to control and acquire SiPM signals. The availability of radiation-tolerant SoCs, combined with the goal of reducing system space and weight, led to the development of an innovative second-generation board, a better-performing device called Cosmo ArduSiPM, suitable for space missions. The architecture of the new detector is based on the Microchip SAMV71 300 MHz, 32-bit ARM^®^ Cortex^®^-M7 (Microchip Technology Inc., Chandler, AZ, USA). While the analog front-end is essentially identical to the ArduSiPM, it utilizes components with the smallest possible package. The board fits in a CubeSat module. Thanks to the compact design, the board has two independent channels, with a total weight of only 40 grams within a CubeSat form factor. The ArduSiPM architecture is based on a single microcontroller and fast discrete analog electronics. It benefits from the continued development of SoCs related to the IoT (Internet of Things) market. Compared with a system with a custom ASIC, this architecture based on software and SoC capabilities offers considerable advantages in terms of cost and development time. The ability to incorporate new commercial SoCs, continuously emerging from advancements in the aerospace and automotive industries, provides the system with a robust foundation for sustained growth over the years. A detailed characterization of the hardware and the system’s response to different photon fluxes is presented in this article. Additionally, coupling the device with a scintillator was tested at the end of this article as a preliminary trial for future measurements, showing potential for further enhancement of the detector’s capabilities.

## 1. Introduction

The necessity to use fragile and big photomultiplier tubes (PMTs) as light detectors limited the use of scintillators in radiation-handled devices. The appearance on the market of new silicon photomultiplier (SiPM) solid-state sensors, with characteristics close to PMT, changes the paradigm [1]. SiPMs are near-single-photon sensors with performance characteristics close to a conventional PMT with the practical advantages of a solid-state sensor and a low-voltage power supply. They are suitable for medical imaging, radiation detection, and bioluminescence applications.

In 2014, at the IEEE NSS/MIC conference in Seattle [2], we introduced the innovative ArduSiPM architecture, which integrated the control and acquisition of a single silicon photomultiplier (SiPM) sensor into one system-on-chip (SoC) microcontroller. This was a major breakthrough, as SiPMs achieved the single-photon limit with a dark count rate (DCR) below 0.05 Mcps/mm^2^ [3] (mega counts per second), eliminating the need for two SiPMs in coincidence to reduce noise. Concurrently, microcontroller manufacturers developed a new generation of SoC chips, enabling the integration of a programmable acquisition system in just a few square centimeters with minimal external components. Our original approach at the time intensively utilized the microcontroller’s peripherals, limiting external components to analog signal conditioning, and its potential was interesting enough that it was subsequently adopted by other research groups as well [4,5,6,7,8,9,10]. The internal counters of the microcontroller functioned as fast event counters and time measurement tools with nanosecond precision, while ADCs and DACs managed analog input and output signals. By knowing the shape of the SiPM signal, we can estimate the number of photons with good accuracy based on the height of the signal with respect to the bias voltage.

The ArduSiPM technology has found applications in many fields of high-energy physics and beyond. The compact size, lightweight, and low energy consumption have identified space applications, particularly for small-sized satellites and CubeSats, as a natural fit for this technology [11]. Thanks to the collaboration with Microchip Technology, a new detector has been developed based on this technology, named Cosmo ArduSiPM, specifically designed for the applications mentioned earlier. Some key performances and design improvements expand its use in a broader range of applications.

The compact design of the Cosmo ArduSiPM detector enables small satellites to integrate it as one of several onboard instruments. This provides an improvement over past missions, where radiation detection equipment, due to its bulk, was often the primary payload [12,13]. When integrated into a small satellite, it can monitor space weather; study the Van Allen radiation belts; detect cosmic rays; and force the hibernation of subsystems at risk from high particle fluxes due to solar flares.

It can also be used to provide information about local gamma-ray radiation background aboard small satellites in space, based on ten-second and minute-period measurements, without the use of a high-voltage Geiger tube. This latter task could also be performed by alternative technologies, either commercial [14] or under development [15].

Additionally, it supports a wide range of scientific studies, such as those in the JEM-EUSO program, which investigates high-energy cosmic rays and other extreme universe phenomena and can contribute to research in astrophysics and environmental monitoring of outer space. The use of such detectors in satellite constellations can be particularly interesting for studying the directions and positions of potential sources of various phenomena.

This type of device can be cost-effective and compete with pricier solutions. Excluding high prototype costs, electronics production costs can be low for a few hundred units. Efficient production estimates place the non-rad-tolerant version under 200 euros and the rad-tolerant version under 2000 euros, excluding commercialization expenses. These costs depend on the current electronics market, affected by post-COVID-19 chip shortages.

## 2. The Cosmo ArduSiPM Architecture

The Cosmo ArduSiPM (Figure 1) is fundamentally based on a similar architecture as its predecessor ArduSiPM [16] and uses pretty much the same Analog circuits. A significant amount of work has been carried out on package reduction aimed at significantly reducing both the size and power consumption of the electronics. The use of a powerful SAMV71 Microchip Technology SoC [17], also available in the radiation-resistant form, gives the possibility to use this device in a radiation-challenging environment such as space [18] or the CERN LHC (Large Hadron Collider) in Geneva accelerator. The Microchip SAMV71 with its ARM^®^ Cortex^®^-M7 CPU core can run at up to 300 MHz, instead of 84 MHz of the SAM38XE ARM^®^ Cortex^®^-M3 CPU (Microchip Technology Inc., Chandler, AZ, USA) core of the original ArduSiPM. The new device revamps the potential of the ArduSiPM with high CPU (Central Processing Unit) processing speed, faster ADCs, faster communication interfaces, and smaller package size. The overall system comprises two main parts: hardware and firmware software.

Cosmo ArduSiPM hardware is a custom-designed printed circuit board (PCB) that incorporates a SiPM interface, an analog front-end, fast data acquisition, a processing system, and several communication interfaces. Photon detection using SiPM, along with its parameters like Time of Arrival (ToA), amplitude, and photon count, requires several signal conditioning stages. The system is made up of a power converter, as a suitable high-voltage DC supply for SiPM; a low-noise voltage amplifier, for better signal-to-noise ratio (SNR); a programmable threshold that controls the fast timing discriminator to cut off the narrow photon pulses and the noise; and a peak hold for ADC (Analog-to-Digital Converter) to read the pulse amplitude accurately. A comprehensive Cosmo ArduSiPM architecture is presented in Figure 2, briefly elucidating the interconnection between the parts.

### Microcontroller Selection

Our strategy for selecting microcontrollers is centered on two key factors: the processing power of the chip and the availability of both analog and digital peripherals crucial for our project’s specific needs. This approach allows us to start with a standard, commercially available chip and, if necessary, upgrade to a version that can handle radiation. Such flexibility is indispensable for the evolving needs of space missions, where both cost and performance are critically balanced.

Commercially available microcontrollers, while cost-effective and well-tested, often fall short of the high standards required for space missions, especially in terms of radiation resistance. Assessing whether these standard chips can endure space conditions is not only a detailed but also an expensive process.

In response to these challenges, we adopted the Microchip SAM S series of microcontrollers, including the Microchip SAMV71 (Figure 3), known for their high-performance processors capable of handling tasks swiftly and precisely. These chips can operate at speeds up to 300 MHz and are equipped with substantial memory capacity, suitable for tasks such as creating histograms for data analysis. Their versatility is further enhanced by various connection options and features that make them adaptable to different applications.

For missions that orbit close to Earth, where radiation levels are comparatively lower, we specifically use the SAMV71Q21RT (Microchip Technology Inc., Chandler, AZ, USA) [19]. This special version of the SAMV71 is designed to resist radiation, making it ideal for such environments. It is immune to latch-up (Single-Event Latch-up (SEL) > 60 MeV) and has undergone rigorous testing for Single-Event Upset (SEU) and Total Ionizing Dose (TID) [20]. This ensures that our devices operate reliably in space without radiation damage, meeting the stringent quality standards required for space equipment.

Furthermore, this radiation-resistant model maintains the same design and pinout as the standard versions, allowing us to use the same printed circuit board (PCB) for different versions of our devices. This uniformity not only simplifies manufacturing but also ensures consistency across our designs. Importantly, the chip is available in small quantities, making it viable for smaller projects and supporting our commitment to cost-effectiveness and scalability.

## 3. Overview of the Discrete Analog Functional Blocks of ArduSiPM

This section provides an overview of the discrete analog functional blocks of the ArduSiPM, external to the SoC. We examine the key components and their roles within the system, emphasizing their importance in enhancing the device’s performance and reliability.

### 3.1. Power Converter

A DC-DC converter provides a low-noise and stable high-voltage DC output to bias the photosensor, with a ripple of less than 0.5 photon equivalent voltage. This power converter is digitally controlled by the microcontroller SoC and can deliver output within the desired range by adjusting the value in a simple resistor network; the standard configuration ranges from 4.5 V to 90 V, starting from a 5 V power source. This setup allows for independent setting of the output span and offset, while an internal DAC enables accurate adjustment of the output voltage with 8-bit resolution within the specified range.

Additionally, an accurate high-side current limit can be set using a resistor to prevent overcurrent damage to the photosensor. An overcurrent alarm is also signaled to the microcontroller SoC if the current exceeds the set limit.

### 3.2. 1-Wire Thermometer

The Cosmo ArduSiPM is equipped with a high-precision 1-wire thermometer, which guarantees an accuracy of 0.5 °C and can be installed close to the photosensor. This thermometer uses a 1-wire interface that requires only one port pin, allowing the temperature to be read as a 9-bit digital value without needing additional components.

At a constant bias voltage, the gain of SiPMs decreases linearly with increasing temperature; this linear variation can negatively impact the photosensor’s performance. Fortunately, these effects can be counteracted by adjusting the bias voltage based on the temperature readings from the thermometer.

### 3.3. Low-Noise Voltage Amplifier

In sensor readout systems, a crucial component is the low-noise amplifier (LNA), which significantly enhances signal quality by amplifying incoming signals while adding minimal noise. This system employs a state-of-the-art commercial discrete operational amplifier with a unity-gain bandwidth of 500 MHz. It has high precision and exhibits low voltage noise, with a voltage offset of less than one millivolt to ensure minimal interference. Unusually, we opted for a cathodic readout of the SiPM, allowing the use of a standard coaxial cable with a single inner conductor for signal transmission. This atypical method is selected for its specific advantage of enabling the SiPM to be positioned at a distance from the board. The primary function of the LNA is to linearly adjust the voltage level of incoming signals, ensuring they meet the input requirements of both the discriminator and the ADC.

Given that the incoming signals can be as short as tens of nanoseconds, it is advisable to employ an amplifier with a swift response to minimize any potential signal degradation. The LNA incorporates a high-performance operational amplifier boasting high input impedance, thus ensuring minimal noise interference within the relevant frequency band of our photodetector. Configured in an inverting setup, this operational amplifier provides the necessary operational gain for optimal system performance (Figure 4). The signal output from the LNA is replicated onto an MMCX (Micro-Miniature Coaxial) connector, allowing precision studies.

### 3.4. Fast Discriminator

We employ one of the fastest single-ended comparators available on the market as a discriminator. It features an impressively quick propagation time of just 7 ns (refer to Figure 5). This comparator excels in detecting pulses that exceed a preset threshold, thus aiding the SAMV71 counter in recording narrow-width pulses. The discriminator can handle the SiPM’s maximum repetition rate, which is in the tens of MHz range, well below the SAMV71 counters’ upper limit of 160 MHz. Moreover, it exhibits a time jitter of less than 1 ns, bettering the 6.6 ns time resolution required for Time of Arrival (ToA) measurements, which depend on the processor’s internal clock. The threshold and width settings of the discriminator are adjustable via the microcontroller, offering flexibility for specific needs. Additionally, the discriminator’s output can be accessed through an output pin, allowing for the external triggering of connected modules or synchronization with other devices, including the second channel of the Cosmo ArduSiPM for achieving hardware coincidence.

### 3.5. Peak Hold

As previously mentioned, the typical SiPM pulse, amplified at the LNA stage, is extremely fast, with a rise time on the order of nanoseconds and a total duration of 20–40 ns, varying by SiPM model. Due to this rapid nature, using the microcontroller’s built-in ADCs to capture the SiPM signal proved challenging; hence, we adopted an unconventional circuit solution for the sensor readout. The SiPM signal is about 100 times faster than the ADCs in cutting-edge microcontrollers. Since capturing the signal with multiple samples was impractical, we implemented a peak-hold circuit equipped with a programmable reset. This circuit extends the duration of the signal to the microsecond level, enabling capture by a 12-bit ADC at 2 Msamples/s. The programmable reset feature allows for fine-tuning the signal in accordance with the specific characteristics of the scintillator and the SiPM, ensuring correct bias restoration of the levels. While this solution increases the dead time on the order of microseconds—compared with the tens of nanoseconds typical of fast SiPM responses—it allows the circuit to handle event rates of up to hundreds of kHz without significant issues (refer to Figure 6).

### 3.6. Communication Interfaces

Fast communication interfaces are essential for such high-speed detectors to timely transfer the measured data to an external computer for postprocessing, storage, or data transmission (for example, for underground data transfer in satellite). Cosmo ArduSiPM takes advantage of copious peripherals available on SoC to offer several industrial high-speed communication interfaces like CANBUS, USB 2.0, UART, and I2C. The Universal Serial Bus (USB) protocol allows communication rates of up to 480 Mbps and enables interfacing with other computers or a wide variety of peripherals. Moreover, there is also a possibility to install a Wi-Fi module to introduce a wireless radio interface to create a network of multiple Cosmic ArduSiPM modules over a large geographical area.

### 3.7. LED Indicators

The Cosmo ArduSiPM boards feature two LEDs, each indicating different conditions. The yellow LED flashes for individual signals above the threshold, while the red LED signals ten pulses within a measurement period. The measurement window is adjustable from microseconds to seconds, with a default period of one second for an event frequency of one per second. During startup, both LEDs blink in a predefined sequence, indicating device readiness and aiding quick verification of operational status and photosensor darkness.

### 3.8. PC/104 PCB Layout

PC/104 stands out as one of the most popular embedded computer standards, known for its smaller, stackable form factors and communication interfaces. With dimensions of 90.17 mm × 95.89 mm, PC/104 boards differ from typical PC motherboards as they are stacked vertically, allowing for space-saving configurations. Thanks to its versatility, this form factor finds extensive use in various applications beyond satellites. In fact, the On-Board Computer (OBC) of a CubeSat often relies on PC/104 technology. Therefore, the layout of the Cosmo ArduSiPM is designed to adhere to the PC/104 standard, ensuring compatibility with a wide range of applications. This design choice enables the integration of two isolated SiPM detectors on a single board, achieved by selecting electronic components with the smallest possible package size. Additionally, for flexibility, the PC/104 board can be divided by cutting it into two separate and fully independent smaller boards, making it suitable for a variety of purposes.

## 4. Assessing Cosmo ArduSiPM: Overview of Electronics Performance, SiPM Noise Characterization, and Response to Various Photon Fluxes

The performance of the Cosmo ArduSiPM was extensively evaluated using two separate test benches: one dedicated to electronic performance and another designed for photonic optical tasks. We also conducted a preliminary gamma spectrometry test with a crystal, intended as an exploratory probe rather than a detailed study. Initially, our focus was exclusively on the device’s electronic characteristics, employing a programmable pulse generator instead of a real SiPM. This setup allowed us to assess the maximum performance achievable, regardless of the sensor model. Subsequently, we used short optical pulses containing just a few photons to operate a Hamamatsu S13660-1325CS SiPM (Hamamatsu, Hamamatsu City, Japan), a small sensor renowned for its very good timing capabilities. We continued our investigations by analyzing the system’s response to various simulated scenarios on an optical test bench. To enhance photon collection, the crystal used in the spectrometry was coupled with a 6 × 6 mm^2^ SiPM.

### 4.1. Electrical Test

#### 4.1.1. The Maximum Counting Rate

The maximum rate for the counter was directly measured using a narrow square pulse with a duty cycle of 50%. We checked the measured rate from Cosmo ArduSiPM counters and, at the same time, the TTL_Out signal from the board and the output of the comparator stage. Compatible with the SAMV71 clock, a maximum frequency of 80 MHz is possible for internal counters. The outcome of the test is a maximum supported rate of 42 MHz. The study of comparator stage output highlights dependence on its setting time for this limit (Figure 7). In the standard operativity of the detector, the maximum measurable rate is affected by the SiPM pulse width, usually greater than 25 ns (40 MHz).

#### 4.1.2. Gain and Front-End Stage Linearity

In these tests, we studied the behavior of the stages that precede the SAMV71 ADC, where the analog signal is digitized: the amplifier and the peak hold. For these measures, we used a 40 ns pulse with variable amplitude to simulate the shape of the pulse compatible with a typical SiPM signal (Figure 5). The maximum signal output of LNA and the voltage level of the peak hold were measured for different amplitudes of the input pulse. As you can see in the diagrams in Figure 8 and Figure 9, the dependence is rather linear for both stages, and the coefficient of determination for the linear regression is very high ($R^{2}$ > 0.9998).

The linear interpolation of data allows one to evaluate the gain (given by the gradient of the line) of the amplification stage, which is 4.48. An offset of a few mV is present in both stages. It is bigger at the output of the peak hold, which by its nature is affected by noise since it locks the maximum noise value. Since the peak-hold output is the signal converted by the ADC, this offset affects the dynamic range of the system. It is possible to compensate for this offset thanks to a 10-bit DAC made available to the SAMV71 analog blocks. The measured value of peak-hold offset can be used to calculate the value for the DAC.

#### 4.1.3. ADC Linearity and Offset Compensation

A similar test was executed to study the SAMV71 ADC linearity. The converted values outputted from the 12-bit ADC, for the different amplitude of input pulse, are reported in the diagram in Figure 10. The analog front-end of SAMV71 has a built-in programmable DAC per channel, allowing it to compensate for the offset. Based on previous measurements, we defined the right value for the Offset Compensation Register. For these measurements, we used the same input signal used for gain and peak-hold stage linearity with the same amplitude steps. The measuring run was repeated twice, with and without offset.

### 4.2. Optical Test

We conducted several tests of the Cosmo ArduSiPM using a dedicated optical test bench at INFN-Roma. For this purpose, we employed a basic setup with a fast LED source positioned in front of the SiPM, within a lightproof box equipped with an optical breadboard. To ensure proper alignment between the SiPM and the LED source, the photosensor is mounted on a 3D-printed custom holder (Figure 11). It is worth noting that the SiPM used has a small area, chosen to maximize temporal performance. While it is possible to use SiPMs with larger areas, in such cases, temporal performance is degraded.

For the test, we chose to use the same type of SiPM usually employed on the standard ArduSiPM [2] application, that is, the MPPC Hamamatsu S13360-1325CS [3]. This photosensor has an area of only 1.3 × 1.3 mm^2^ and a pitch of 25 μm with a fill factor of 47%. The typical dark count at the standard ambient temperature (25 °C) is only 70 kcps; the low rate of dark count allows us to use a single channel in standard scintillation detectors, reducing the noise to fractions of cps with a threshold of just 3 photoelectrons;

The light source used in this arrangement is an LED driver Picoquant PDL 800-B equipped with a sub-nano-second pulsed LED head. This source can generate a fast light pulse at 800 ps (FWHM) with a wavelength of 450 nm and a fine adjustable optical power until 2800 μW peak [21]. It is possible to set the amplitude of every LED pulse to work in the single photoelectron or few photoelectron regimes; this is important to emulate the standard condition of working of SiPM in Cosmo ArduSiPM-based particle detectors. The maximum rate of the LED driver is 80 MHz, which permits the measurement of the maximum acquisition rate of the Cosmo ArduSiPM system in a typical configuration.

Given the limited range of selectable frequencies on the PicoQuant LED driver, the repetition rate of the light pulse was controlled using an external TTL pulse generator capable of selecting frequencies ranging from 0 to 40 MHz. To ensure an entirely dark environment, the lightproof box with source and sensor is wrapped in a black nonwoven fabric. The dark count condition was verified before each measurement.

#### 4.2.1. SiPM Dark Count Evaluation and Dependency of the Threshold

The dark count is an important feature of SiPM. Due to the discrete nature of the SiPM signal, the switching on of each SiPM cell gives an additional and quantifiable contribution to output signal amplitude. This effect is magnified thanks to the LNA amplification.

Based on Cosmo ArduSiPM applications, it is very useful to know the expected dark count rate as a function of the selected threshold, as well as the number of SiPM cells that must switch on simultaneously to reach the overthreshold condition. To study this dependence, the SiPM was placed inside the lightproof box, at room temperature (25 °C), without any light source [22]. A custom Python 3.7 script adjusts the 12-bit DAC value within the range of 0 to 1000 while measuring the time required to reach at least 400 threshold-crossing events. For threshold values where the dark count rate exceeds 400 Hz, the measurement duration is set to 1 s; this approach ensures the attainment of a statistically substantial sample size maintaining a measurement error below 5%. The process aims to determine the repetition rate of dark counts. The last value of 1000 for the DAC was chosen based on the time required to reach 400 events that grow with the threshold. In Figure 12, the result of this measurement is presented.

As is evident, there is a clear separation between dark count events switching on one or more cells of SiPM. Whenever the threshold becomes greater than the typical amplitude of one single cell signal, we can see a steep decrease in the counting rate, followed by a flat zone corresponding to a double amplitude of the signal. The separation is quite constant for each additional cell.

This measurement facilitates the selection of a threshold value within the plateau region between consecutive inflection points, enabling the identification of events associated with the simultaneous activation of a specific number of SiPM cells, thereby achieving stability against gain fluctuations. Thanks to this procedure, we can easily characterize the dark count of the utilized SiPM and determine the optimal threshold based on the application requirements. Additionally, if necessary, we can subtract the expected dark count from the final rate measurements. For example, in bioluminescence applications, where we attend to a high number of photons with a flat time distribution, we can use a “single cell” (i.e., a single p.e.) threshold to measure the increase in overthreshold events rate [23,24]. Moreover, as a cosmic ray counter, where we expect a low rate of events yielding a lot of photons inside the scintillator, a threshold that allows one to overlook the dark count rate is preferable. In particular, the presence of a scintillator coupled with the SiPM would cause the count rate to be dominated by the cosmic rays rate for thresholds greater than three SiPM-lit cells (i.e., three photoelectrons).

#### 4.2.2. The Maximum Counting Rate in the Optical Bench

The maximum possible rate for the Cosmo ArduSiPM readout is evaluated in a similar way used for electrical tests. This measurement takes into account the capacity of SiPM and the limit due to pulse width. To simulate a typical scenario, we use a pulse consisting of a few photons. Due to the stochastic nature of LED light emission, there is some variability in the number of photons per pulse, which was set to an average of eight to nine photons. This is achieved by manipulating the solid angle as viewed by the detector in relation to the source, taking advantage of the fact that the source has a specific beam angle and the SiPM (silicon photomultiplier) intercepts a fraction of this solid angle.

Until 40 MHz, the Cosmo ArduSiPM is able to measure the pulse rate without errors. Since we used a 20 ns width pulse in the electrical test, we expected this result, even if the SiPM signal has a longer tail.

#### 4.2.3. ADC Signal Amplitude Spectrum

The SAMV71 microcontroller is equipped with a configurable 12-bit ADC that allows measurements of signal amplitude thanks to the Cosmo ArduSiPM peak hold (Section 3.5). In our configuration, the maximum achievable resolution for the ADC is 0.2 mV. Since the SiPM can only deliver pulses with amplitude levels that are multiples of the amplitude generated by a single switching-on cell, if the resolution is sufficient to distinguish between two adjacent amplitude levels, it means we can accurately determine the number of cells that have switched on and thus extract all the available information from the pulse amplitude.

In this context, we investigated the system’s ability to distinguish between two consecutive peaks of the amplitude histogram using the same LED source settings. The obtained results are compared with those from an analog study conducted using a 12-bit, 400 MHz, 2 GS/s oscilloscope Lecroy WaveRunner HRO 64 Zi (Figure 13). Despite the significant difference in acquisition numbers, primarily caused by the longer dead time of the oscilloscope compared with our system, both devices yield a similar measurement of approximately 3 mV between consecutive peaks of the amplitude histogram.

The analog signal was picked up from Cosmo ArduSiPM analog output, showing the SiPM signal after the first amplification stage (Figure 14).

Like every ArduSiPM detector, Cosmo ArduSiPM measures the time between consecutive signals as the number of the MCU (Microcontroller Unit) clock pulses; this means 6.6 ns is the limit for the SAMV71 microcontroller family. The measured jitter between the LED pulse and the TTL output of the Cosmo ArduSiPM is around 600 ps, around one order of magnitude lower than the highest possible resolution.

#### 4.2.4. Comparison between Threshold Scan Technique and ADC Measurements for Few Photon Light Signals

This same setup (Figure 15) was used to simultaneously perform a threshold scan and an ADC amplitude sampling. The idea is again to carry out two independent amplitude measurements and see if the gain corresponds. The way to extract the information on peak amplitude from a threshold scan can be visualized as a derivation (Figure 16). In this way, we know that the points of flex of the threshold scan will correspond to the maxima of the p.e. amplitude peaks.

The threshold scan is performed as in Section 4.2.1, but this time the LED pulse is on (although of 1–2 photons). The amplitude sampling is performed as just seen. The curve smoothing and peak location are performed numerically via a Python script. Results are shown in Figure 17, where both analyses report a gain of about 2.7 mV. The gain is slightly lower than the one registered before because the LED pulses used in this case were lower than those in Figure 13.

## 5. Preliminary Test of the System as Spectrometer

Once the performance of Cosmo ArduSiPM electronics is detailed, we can try to observe the device being assembled into a scintillation detector. Specifically, we provide an overview of its potential as a gamma spectrometer.

The readout system we just characterized is composed of a SiPM Hamamatsu 13360-6025CS (Hamamatsu, Hamamatsu City, Japan) which, thanks to its 6 × 6 mm^2^ surface, allows for a broader light collection. The SiPM is then coupled to a BrilLanCe^®^380 scintillator (Saint-Gobain Ceramics & Plastics, Inc., St. Pierre less Nemours, France) [25] (Cerium-doped Lanthanum Bromide) and exposed to a ^60^Co source, as shown in Figure 18.

^60^Co is an excellent calibration source because it undergoes beta decay with a maximum energy of 317.9 keV and emits gamma rays with well-defined energies of 1173.2 keV and 1332.5 keV. These distinct and well-characterized energy peaks provide precise reference points for calibration.

LaBr_3_(Ce) has a high radiation yield and response times suitable for the chosen SiPM without any special adjustments.

The acquisition results denote the detector’s ability to function as a gamma spectrometer [26].

Referring to Figure 19, we distinguish, from left to right, the 209.8 keV backscattering peak, the 963.4 keV Compton edge, the 1173.2 keV first gamma peak, the 1332.5 keV second gamma peak, and the 2614 keV peak, which is the sum of the previous two. By fitting the two gamma peaks with a double Gaussian, we can obtain the mean and deviation of the measurement, performing an energy calibration for this SiPM–scintillator coupling. With these data, we calculate two “ADC step to keV” scale conversion factors and choose to use the average of the two as the reliable value. For the configuration 6025CS SiPM–LaBr_3_(Ce) scintillator, the ^60^Co calibration gives us 1keV=1.82±0.03ADCsteps.

## 6. Conclusions

ArduSiPM technology has become a significant application in the SoC (system-on-chip) industry for all-in-one detectors, renowned for integrating processors, memory, and peripherals on a single chip. Our recent evaluation of the Cosmo ArduSiPM SoC’s second generation of ArduSiPM technology has affirmed its exceptional versatility and robust performance.

We emphasize the importance of firmware for its flexibility, allowing us to upgrade whenever we find better algorithms. This ensures continuous improvement of the system, including peripherals, interrupts, memory, and external communication. This approach consistently enhances detector performance without changing the hardware. Despite its complexity, these improvements are essential for the ongoing development and efficiency of the device.

Our key objective is to enable the Cosmo ArduSiPM SoC to autonomously generate histograms and perform reduction, thus reducing reliance on external computing. This move makes processing more efficient and positions the Cosmo ArduSiPM at the forefront of detectors with edge computing capabilities. 

## Figures and Tables

**Figure 1 sensors-24-03836-f001:**
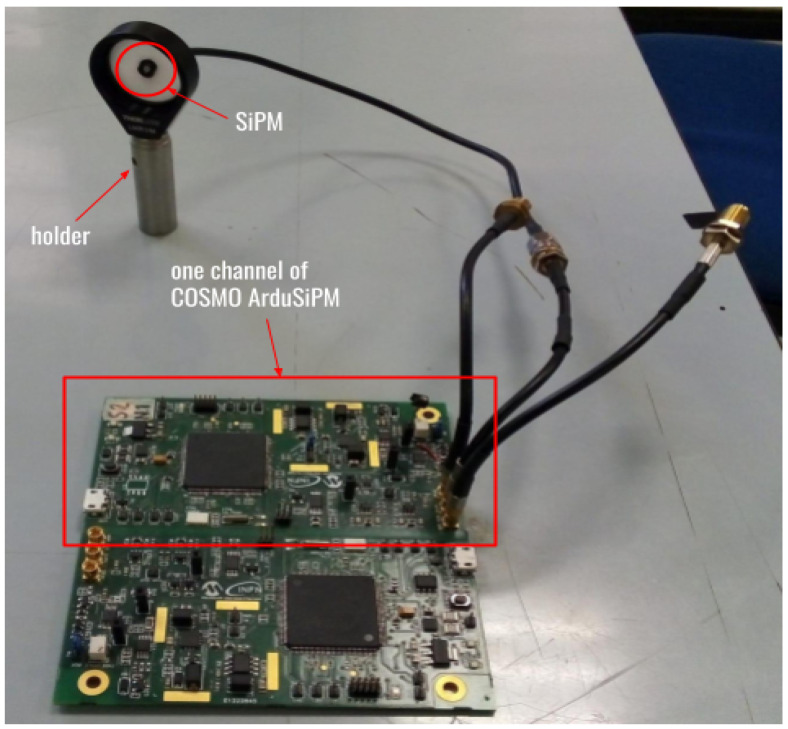
Picture of the Cosmo ArduSiPM board linked to the SiPM placed on a holder. Each board is composed of two independent channels.

**Figure 2 sensors-24-03836-f002:**
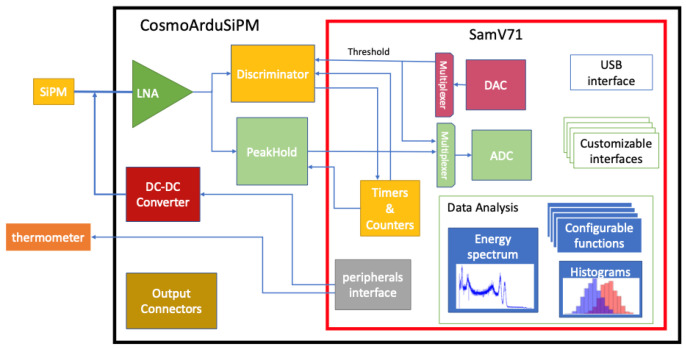
Block diagram of Cosmo ArduSiPM. The implementation of most functions relies on the microcontroller’s internal peripherals. Depending on the specific application, the analysis and interfaces can be configured accordingly.

**Figure 3 sensors-24-03836-f003:**
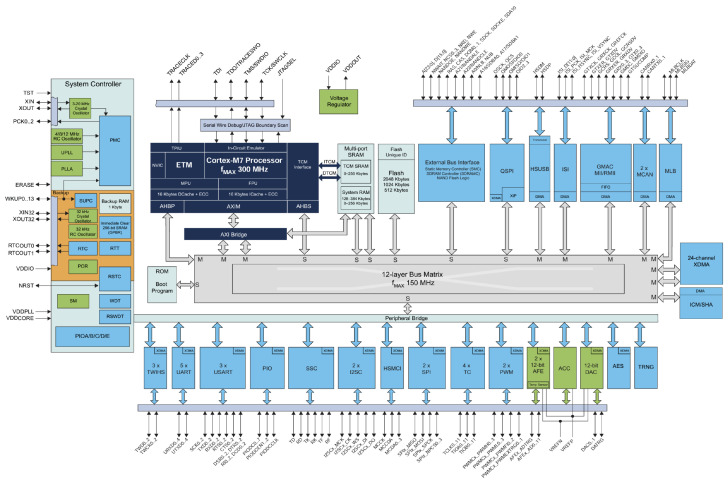
Architecture of the SAM V71 family.

**Figure 4 sensors-24-03836-f004:**
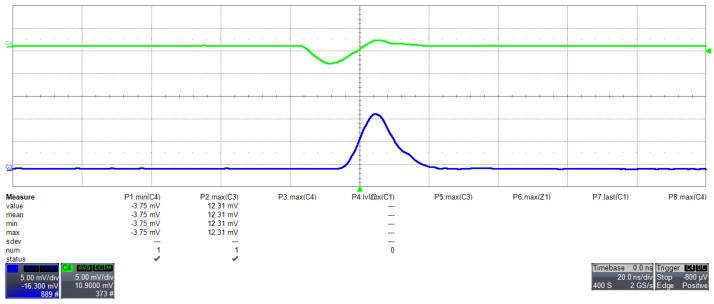
Response of the LNA to an injected signal pulse studied using a 12-bit oscilloscope Lecroy WaveRunner HRO 64 Zi (LeCroy Corporation, Chestnut Ridge, NY, USA). The green waveform represents the input signal from a pulse generator. The pulse is 1 mV, with a width of 8 ns and a transient time of 5 ns, closely resembling a few-photon pulse from a SiPM. The blue signal is the output of the LNA in the standard configuration of the Cosmo ArduSiPM.

**Figure 5 sensors-24-03836-f005:**
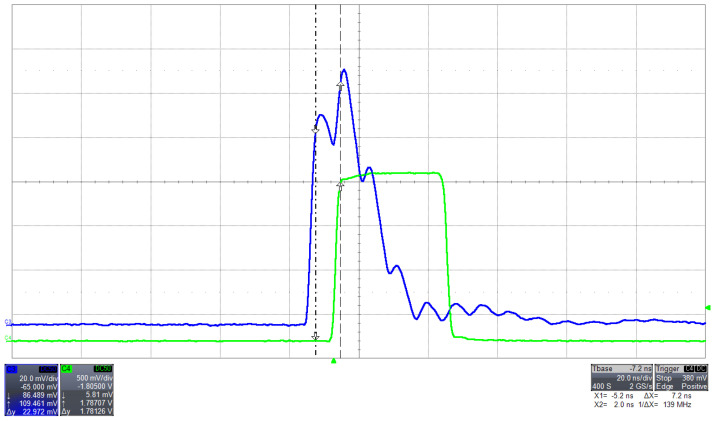
In blue, the pulse from a SiPM Hamamatsu 13360-1325CS acquired with the oscilloscope (20 mv and 20 ns per div.) from the analog output of Cosmo ArduSiPM. We measure a 7 ns delay between the SiPM pulse and the output of the comparator (TTL_out, green). The multiple peaks are due to cable reflections.

**Figure 6 sensors-24-03836-f006:**
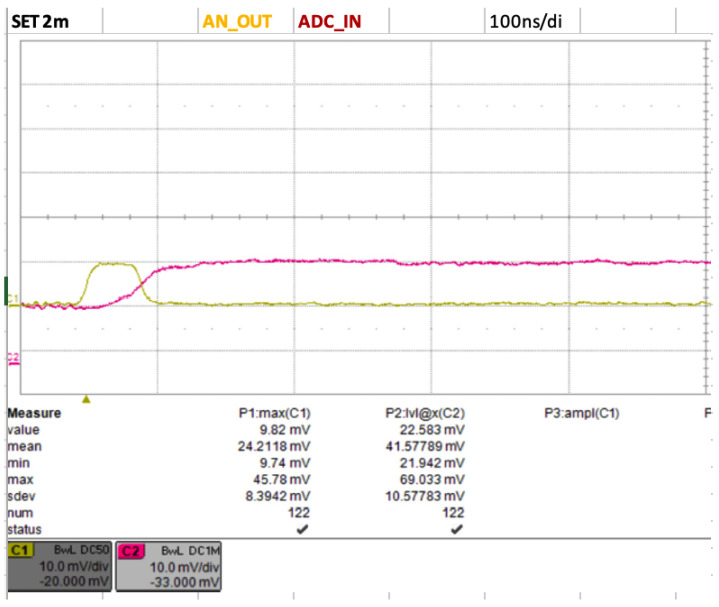
The output of the peak-hold circuit (fuchsia signal) generated by a pulse signal of few ns (yellow signal). The waveforms are acquired with a oscilloscope Lecroy WaveRunner HRO 64 Zi. The peak hold keeps the voltage level at the maximum of the input signal without appreciable discharge. The time scale is 100 ns/division.

**Figure 7 sensors-24-03836-f007:**
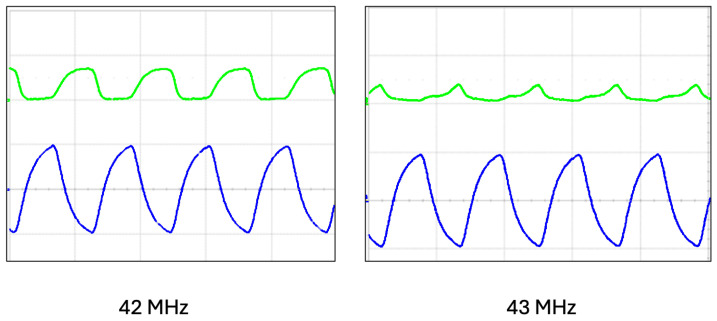
Comparator output and analog output for 42 MHz (left picture) and 43 MHz (right picture) of square pulse input. The blue signal is the output of the LNA stage (amplitude scale 100 mV/div) and the green one is the square TTL output from the comparator (amplitude scale 5 V/div) The time scale of the horizontal grid was 20 ns/div.

**Figure 8 sensors-24-03836-f008:**
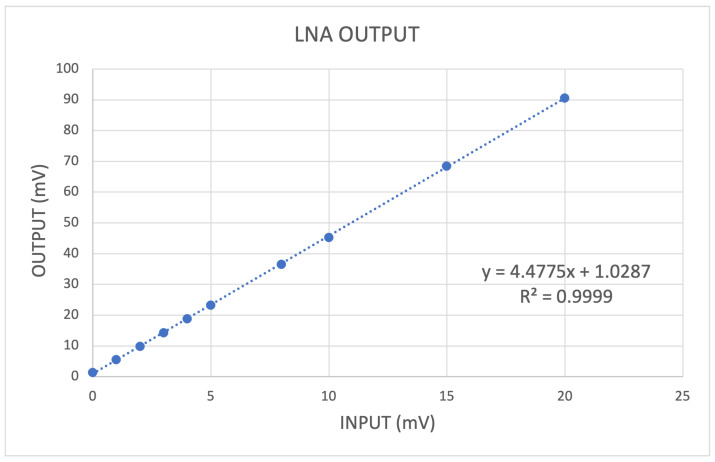
Linear regression of LNA output as a function of the signal input amplitude. A fast 40 ns signal pulse, with 5 ns of transient time, is used as input to simulate a typical SiPM pulse.

**Figure 9 sensors-24-03836-f009:**
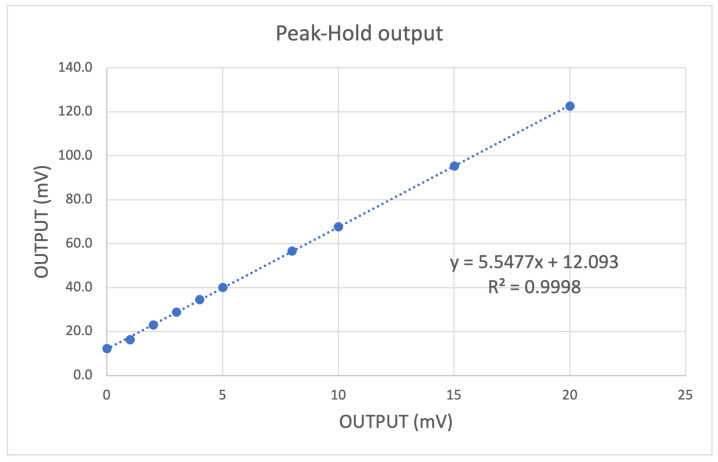
Linear regression of peak hold output as a function of signal input amplitude. A fast 40 ns signal pulse, with 5 ns of transient time, is used as input to simulate a typical SiPM pulse. The offset is a feature of the peak hold circuit that depends on each individual device and can be compensated.

**Figure 10 sensors-24-03836-f010:**
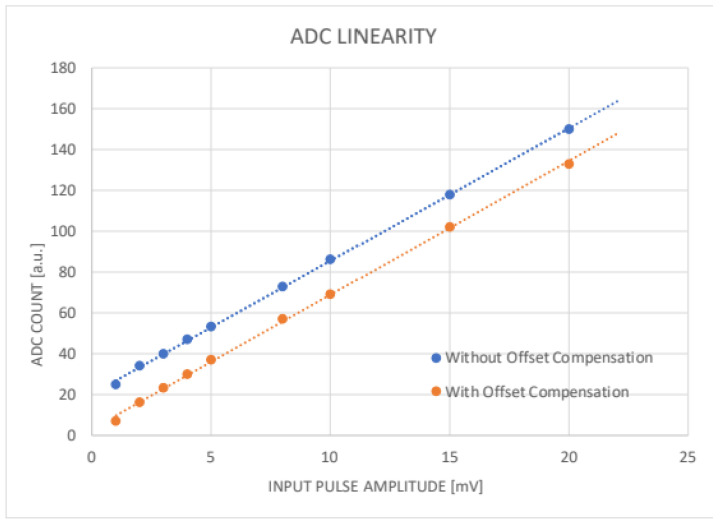
Linear regression of ADC digital output as a function of signal input amplitude. A fast 40 ns signal pulse, with 5 ns of transient time, is used as input to simulate a typical SiPM pulse. The curves with and without the right offset compensation value are shown.

**Figure 11 sensors-24-03836-f011:**
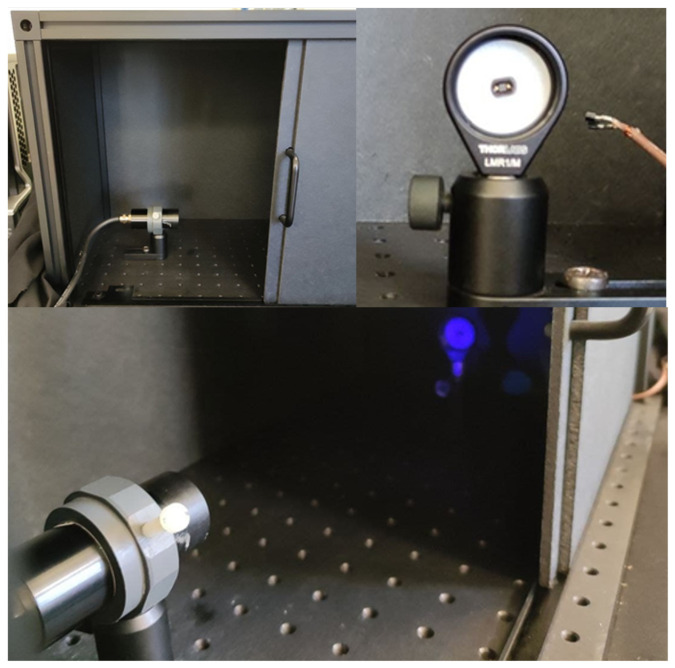
Experimental setup for optical test, including LED head and SiPM placed in the lightproof box. Below, the LED light reflecting on the white holder.

**Figure 12 sensors-24-03836-f012:**
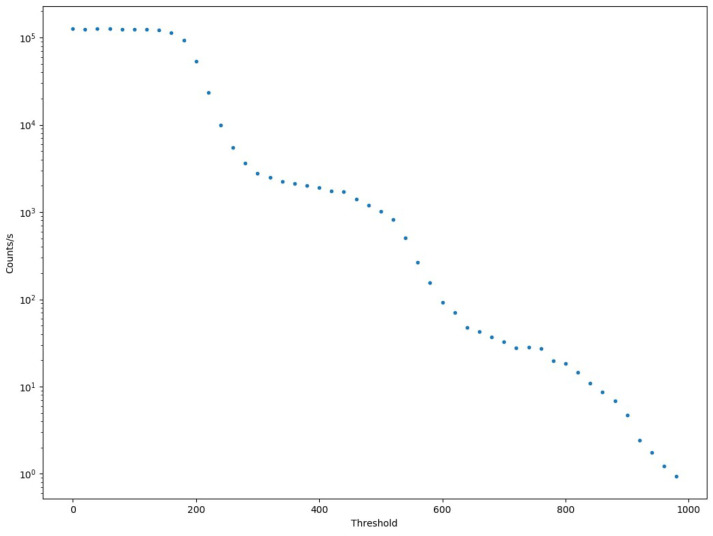
Threshold scan of the used SiPM in dark conditions. The ladder structure gives reliable results on dark frequency for each coincidence count.

**Figure 13 sensors-24-03836-f013:**
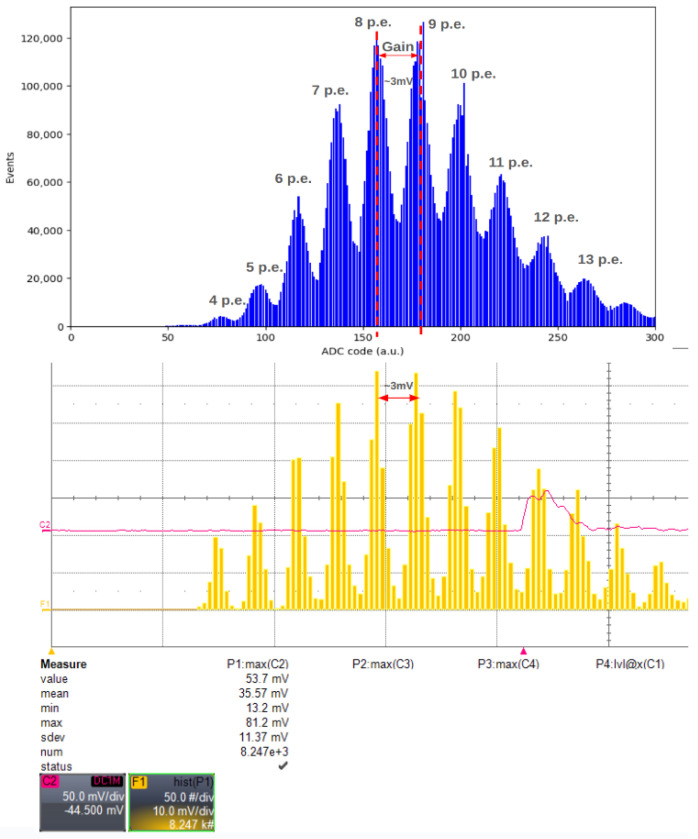
Comparison between amplitude histogram produced with the Cosmo (**top**) and the LeCroy Oscilloscope (**bottom**). The gain is defined as the distance between two consecutive peaks.

**Figure 14 sensors-24-03836-f014:**
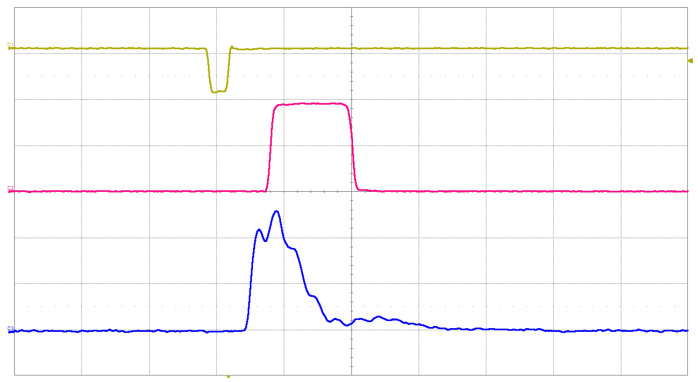
In this figure, we show the LED source driver signal (yellow, amplitude scale 1 V/div), TTL output from Cosmo ArduSiPM (magenta, amplitude scale 1 V/div), and amplified SiPM analog signal (blue, amplitude scale 5 mV/div) captured by a 400 MHz bandwidth oscilloscope. The time scale of the horizontal grid was 20 ns/div.

**Figure 15 sensors-24-03836-f015:**
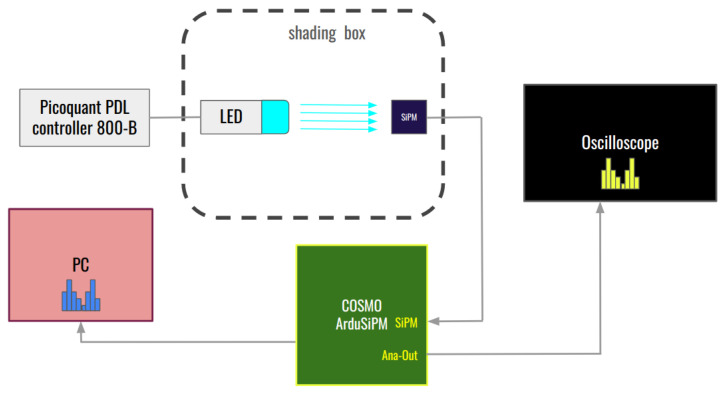
Block diagram of the Cosmo ArduSiPM optical test bench for amplitude measurements.

**Figure 16 sensors-24-03836-f016:**
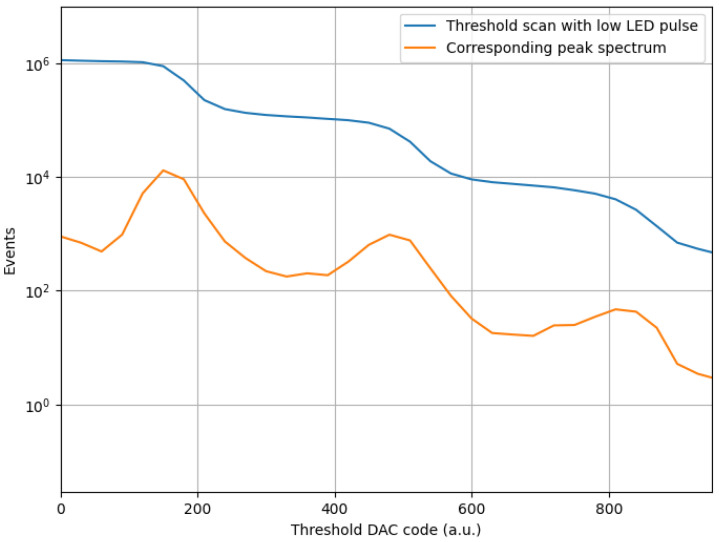
Threshold scan performed with the LED on (blue) and corresponding numerical derivation (orange). We can see how this operation returns amplitude peaks.

**Figure 17 sensors-24-03836-f017:**
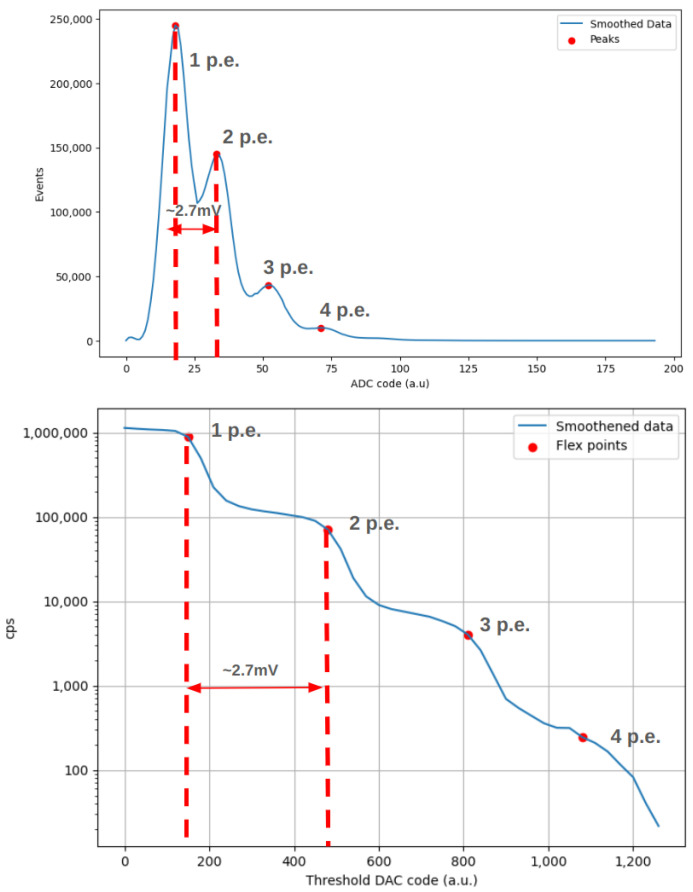
Comparison between smoothened histogram produced by Cosmo ADC and threshold scan from Cosmo Threshold Dac.

**Figure 18 sensors-24-03836-f018:**
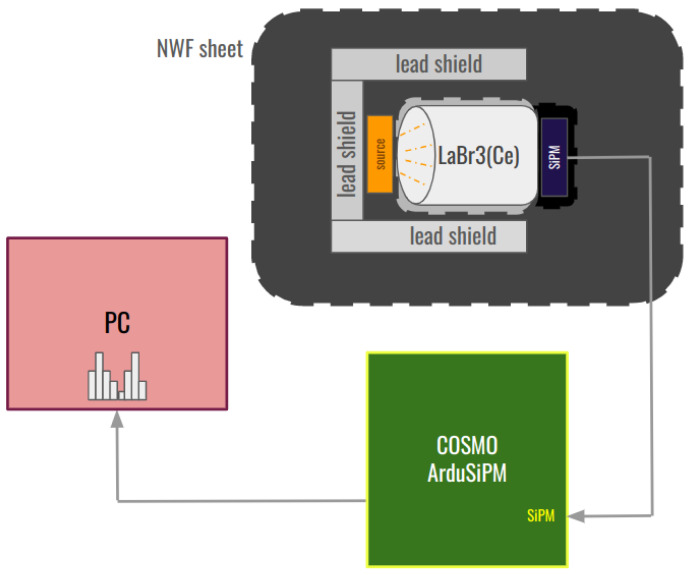
Block diagram of the Cosmo ArduSiPM test bench for spectrometric measurements.

**Figure 19 sensors-24-03836-f019:**
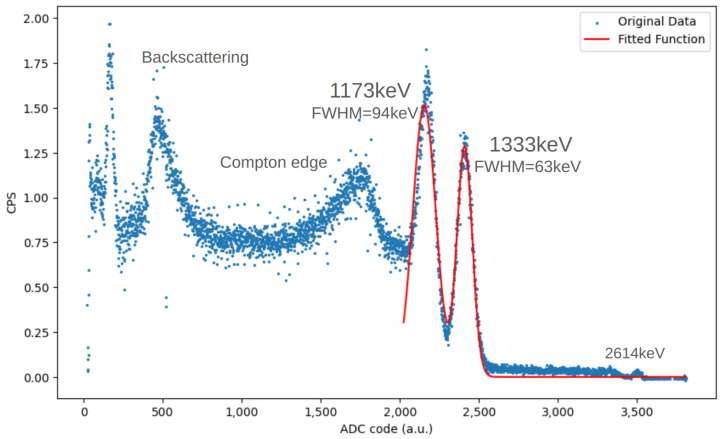
^60^Co gamma spectrum obtained with Cosmo ArduSiPM. The characteristic peaks at 1173 keV and 1333 keV are easily detectable.

## Data Availability

Data are contained within the article.

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
