# Peer review of "Cosmo ArduSiPM: An All-in-One Scintillation-Based Particle Detector for Earth and Space Application"

_sensors, 2024, doi:10.3390/s24123836_

Round 1

Reviewer 1 Report

Comments and Suggestions for Authors

Dear Authors,

Thank you for submitting your manuscript titled "Cosmo ArduSiPM: An All-in-One Scintillation-Based Particle Detector for Earth and Space application" to Sensors. The integration of SiPM sensors and SoC technology to create the Cosmo ArduSiPM represents a significant advancement in particle detection technology. The detailed description of the microcontroller's use in signal acquisition and control, along with the architectural improvements for space applications, is particularly impressive. The design's focus on reducing space and weight while maintaining performance within a CubeSat module is interesting.

While the paper is very interesting and informative, a minor revision is recommended to enhance clarity in some sections. Specific comments and suggestions will follow this note. 

I look forward to the revised manuscript.

Kind regards,

Reviewer.

General comments.

To make the manuscript easier to read and understand, it would be helpful to add a list of all abbreviations at the end. This will improve the paper's flow and create a central reference for all the abbreviations used. It's important to note that the same abbreviations can have different meanings in various scientific fields. Including this list will also assist readers in comprehending the content more easily. This addition will enhance the overall quality of the manuscript. 

Could you please ensure that the dimensions of the units are consistently separated by a space throughout the entire manuscript? This will help maintain clarity and consistency for the readers (for example see caption for figure 4 and text in line 144). 

If it's possible, could you please include rough cost estimates in the manuscript for the system used? This would help anyone interested in building a similar device for different applications, such as environmental remediation of nuclear legacy waste, to understand the approximate funds needed for the project. This information would be greatly beneficial for readers looking to replicate the system.

Specific comments.

Line 93.

Please provide a citation or reference where one can find more general information about Single Event Upset and Total Ionizing Dose testing for electronic devices.

Line 139.

Could you please remove the extra period in "performance.(Figure 4)"? 

Line 189.

Could you please include the dimensions as SI units (in brackets) for clarity?

Lines 257, 258.

Could you please ensure that the unit dimension for the dark count is consistent both in the unit itself and in the explanation provided in the brackets?

Line 281.

Could you please add a space in front of the bracket?

Line 303.

Please add definition of 'p.e.' to the list of abbreviations.

Figure 19.

Could you please provide the full width at half maximum (FWHM) information for the Co-60 gamma peaks displayed in the figure?

Author Response

Dear Reviewer,

We sincerely thank you for your invaluable suggestions and for meticulously identifying the corrections in our manuscript. We greatly appreciate your insightful feedback. We are pleased to inform you that all the suggested corrections have been implemented.

Kind regards,

Valerio Bocci

Reviewer 2 Report

Comments and Suggestions for Authors

I have carefully read this paper, which describes an all-in-one particle detector based on Arduino. The paper, overall, is interesting, but suffers from the low quality of its English form, which, sometimes, makes it difficult to read. Also, the paper seems to be written in a hurry, sometimes the units of measure are in italics, digits and units are not separated by a space, and the quality of some figures is well below average. In many points the sentences are unclear. Therefore, before going into details about the scientific merit of this paper, I kindly ask the authors to re-write it in a clear, concise, English-corrected form; a revision by an English mother speaker, expert of tecnhical language, would be quite advisable. I have marked on the attached file some corrections to the paper, but they should be only considered as a starting point for a much more extensive revision. For what concerns the paper content, it basically focussed on electronics, while more results about the performance of the system, like the ones presented in Section 5, would be very welcome. I would suggest the authors to improve this part.

Comments on the Quality of English Language

I have carefully read this paper, which describes an all-in-one particle detector based on Arduino. The paper, overall, is interesting, but suffers from the low quality of its English form, which, sometimes, makes it difficult to read. Also, the paper seems to be written in a hurry, sometimes the units of measure are in italics, digits and units are not separated by a space, and the quality of some figures is well below average. In many points the sentences are unclear. Therefore, before going into details about the scientific merit of this paper, I kindly ask the authors to re-write it in a clear, concise, English-corrected form; a revision by an English mother speaker, expert of tecnhical language, would be quite advisable. I have marked on the attached file some corrections to the paper, but they should be only considered as a starting point for a much more extensive revision. For what concerns the paper content, it basically focussed on electronics, while more results about the performance of the system, like the ones presented in Section 5, would be very welcome. I would suggest the authors to improve this part.

Author Response

Dear Reviewer,

We would like to extend our sincere gratitude for your valuable insights and suggestions. Your feedback has sparked significant discussions within our group, leading to enhanced clarity and improvements in our work.

In response to your requests:

  1. Space Applications Enabled by the Device: We have described the various space applications enabled by our device, surveying and citing the use of similar devices in space missions.

  2. Improvements: We have provided information on the limitations in speed, accuracy, and resolution that our new device has addressed.

  3. Circuit Details and Innovations: We have attempted to provide further details on the circuits. However, we are unable to disclose additional information as these details are likely subject to a technology transfer in the near future.

Thank you once again for your constructive feedback.

Best regards,

Valerio Bocci

Reviewer 3 Report

Comments and Suggestions for Authors

Please describe the space applications enabled by the device and survey the uses of this type of device in current or previous space applications. Please also give some information about limitations in speed, accuracy and resolution that have been improved specifically for space applications by the new device.

Some analog functionality is described insection 3.3,3.4, and 3.5. However, no details of the circuits and innovations at the circuit level is given. Please include a more detailed description of the circuits, and not just an input output description.

Comments on the Quality of English Language

N/A

Author Response

Dear Reviewer,

We sincerely thank you for your invaluable suggestions and for identifying the necessary corrections in our manuscript. We greatly appreciate your insightful feedback.

In response to your requests:

  • We have corrected all the errors you highlighted.
  • We have included the new measurements in the abstract.
  • Regarding the scintillator, we have expanded the paragraph to provide a preliminary overview. While the article could have concluded without the scintillator measurement, having data that demonstrate its functionality, we preferred to include these initial measurements as an interesting preview of a future article dedicated to this topic.

Thank you once again for your constructive feedback.

Valerio Bocci

Reviewer 4 Report

Comments and Suggestions for Authors

See the attached report.

Comments on the Quality of English Language

In general very good; only few changes and clarifications needed. See the attached report. 

Author Response

Dear Reviewer,

We deeply appreciate your thoughtful feedback and suggestions. Your contributions have ignited meaningful discussions among our team members, significantly improving our work's clarity and quality. In response to your comments:

  • References to Related Work: We have conducted a thorough search and added citations to other groups that have pursued similar lines of research or have used slightly different approaches.

  • Circuit Details: We have attempted to provide additional details on the circuits. However, we are unable to disclose more information as these details are likely subject to a technology transfer in the near future.

  • Implementation of Other Suggestions: All other suggestions have been incorporated into the manuscript.

Best Regards,

Valerio Bocci

Reviewer 5 Report

Comments and Suggestions for Authors

The paper describes the development and testing of a second-generation Arduino Due-based System on Chip (SoC) board, called Cosmo ArduSiPM, for the readout of silicon photomultiplier sensors (SiPM) for operation in space applications on board a CubeSat module.

To the best of my knowledge, the work is novel and original. I think that the title of the paper is appropriate. In the abstract, the work is well motivated, the approach is comprehensively summarized, and the main results are highlighted. The introduction should be improved, as it circles too much around the authors’ own works. The necessity and relevance of the work is well motivated, but it is not put into proper context to prior publications of other groups. The sections on the Cosmo ArduSiPM architecture and on its discrete analog functional blocks are well comprehensible, just some information on the selected components is missing. The section on the performance of the Cosmo ArduSiPM extends over a total of 9 pages, albeit it is being called “a brief overview”. In contrast, the section on spectrometric measurements is much too short. I missed at least a mention of the energy calibration of the system. In the conclusion, the results are briefly summarized, and a short outlook on future software developments is given.

In revision, the following points have to be addressed:

1)      Introduction: the work should be put into proper context to previously published work of other group. Only three papers are cited in the introduction, two of which are the authors’ own. Of all 13 references cited throughout the paper, 7 are the authors’ own work, 4 are datasheets, and only 2 are scientific publications of others. Please cite relevant paper of other groups and relate their findings to your work.

2)      Figures 2 and 3 are not cited in the text. Please refer to them in your manuscript.

3)      I think Figure 3 is too small to be well legible. I suggest to enhance it by a factor of approximately 1.5, that would still fit the page well.

4)      Section 3: the part numbers and manufacturers of the key components external to the SoC should be added, for instance of the DC-DC converter, the 1-wire thermometer, the low-noise amplifier, etc.

5)      Figure 7: axis titles and labels are missing. Please mention at least in the caption the time per division of the horizontal grid, and the voltage per division of the vertical grid.

6)      Figures 8 and 9: I think the axis labels have been accidentally mixed up: I assume the x axes of both graphs should be labeled “Input (mV)” and the y axes should be “Output (mV)”. Please check and correct.

7)      Figure 14: axis titles and labels are missing. Please mention at least in the caption the time per division of the horizontal grid, and the voltage per division of the vertical grid.

8)      Figure 17: please explain the abbreviation “p.e.” in the caption.

9)      Section 5 on Spectrometric measurements: the section is extremely short. I suggest to add at least a brief discussion on the performance and on the calibration (i.e. conversion factor from “ADC code” to energy scale in keV).  

In addition, I noted the following typos:

10)  Line 218 and Figure 7: “MHz”

11)  Lines 112, 221, 225, 263, 264, 271, 319, captions of Figures 4, 7, 10 and 19: add spaces between numbers and units of physical quantities.

12)  “Conclusions” section: at 4 occurrences, the acronym “ArduSiPM” is misspelled with respect to capitalization of letters.

13)  Line 353: “through a Python script”, or (better) “via a Python script”.

14)  Reference [13]: the numbering is double.

Author Response

Dear Reviewer,

We are sincerely grateful for your valuable comments and suggestions. Your observations have sparked significant discussions within our team, greatly enhancing the clarity and quality of our work. Below, you will find our responses to your observations.

References to related works: We have conducted thorough research and added citations to other groups that have pursued similar research lines or adopted slightly different approaches.

Details on circuits: We have attempted to provide additional details on the circuits. However, we cannot disclose further information as these details will likely be subject to a technology transfer in the near future.

Implementation of other suggestions: All other suggestions have been incorporated into the manuscript.

Best Regards

Valerio Bocci

Round 2

Reviewer 2 Report

Comments and Suggestions for Authors

The authors have improved the paper according to the suggestions of the referee doing, overall, a very good work. I suggest that this paper is published as it is.

Reviewer 3 Report

Comments and Suggestions for Authors

The paper is improved from the last revision. It is ready for publication.